# Regulation of chromatin architecture by transcription factor binding

**Stephanie Portillo-Ledesma[1,2], Suckwoo Chung[1], Jill Hoffman[1], Tamar Schlick[1,2,3,4]***

[1]Department of Chemistry, 100 Washington Square East, Silver Building, New York University, New York, United States; [2]Simons Center for Computational Physical Chemistry, 24 Waverly Place, Silver Building, New York University, New York, United States; [3]Courant Institute of Mathematical Sciences, New York University, New York, United States; [4]New York University-East China Normal University Center for Computational Chemistry, New York University Shanghai, Shanghai, China

**\*For correspondence:**
schlick@nyu.edu

**Competing interest:** The authors declare that no competing interests exist.

**Abstract** Transcription factors (TF) bind to chromatin and regulate the expression of genes. The pair Myc:Max binds to E-box regulatory DNA elements throughout the genome to control the transcription of a large group of specific genes. We introduce an implicit modeling protocol for Myc:Max binding to mesoscale chromatin fibers at nucleosome resolution to determine TF effect on chromatin architecture and shed light into its mechanism of gene regulation. We first bind Myc:Max to different chromatin locations and show how it can direct fiber folding and formation of micro-domains, and how this depends on the linker DNA length. Second, by simulating increasing concentrations of Myc:Max binding to fibers that differ in the DNA linker length, linker histone density, and acetylation levels, we assess the interplay between Myc:Max and other chromatin internal parameters. Third, we study the mechanism of gene silencing by Myc:Max binding to the Eed gene loci. Overall, our results show how chromatin architecture can be regulated by TF binding. The position of TF binding dictates the formation of microdomains that appear visible only at the ensemble level. At the same time, the level of linker histone and tail acetylation, or different linker DNA lengths, regulates the concentration-dependent effect of TF binding. Furthermore, we show how TF binding can repress gene expression by increasing fiber folding motifs that help compact and occlude the promoter region. Importantly, this effect can be reversed by increasing linker histone density. Overall, these results shed light on the epigenetic control of the genome dictated by TF binding.

## eLife assessment

In this **important** study, chromatin is simulated as a polymer at the scale of genes, and the 3D organization of chromatin is analyzed at nucleosome resolution. There is **convincing** evidence for the emergence of chromatin microdomains due to the action of transcription factors, based on the simulation incorporating well-known biophysical properties of DNA, of nucleosomes, of linker histones, and of the transcription factor pair Myc:Max, as well as considering how the 3D organization of chromatin results from bending and looping of DNA. The work greatly improves our understanding of how the joint action of transcription factors and chromatin features affects chromatin structure and accessibility, which is of interest to anyone studying gene regulation.

## Introduction

Eukaryotic genomes are compactly packaged inside the cell nucleus by wrapping ~147 bp of DNA around an octamer of histone proteins. This first level of organization establishes the nucleosome, the chromatin basic repeating unit (*Luger et al., 1997*; *Kornberg, 1977*; *Onufriev and Schiessel, 2019*).

Chains of nucleosomes form chromatin fibers that undergo additional folding that increases compaction. This folding can be aided by the binding of proteins. For example, CTCF and cohesin execute loop extrusion to form Topologically Associated Domains or TADs (*Rowley and Corces, 2018*). Linker histones (LH) bind to the nucleosome at the entry/exit DNA sites to compact chromatin fibers and regulate their architecture (*Portillo-Ledesma et al., 2022*; *Perišić et al., 2019*). Finally, TFs bind to regulatory DNA elements, modulating chromatin architecture and accessibility and, thereby gene expression (*Weidemüller et al., 2021*).

Myc is a TF with a C-terminal basic helix-loop-helix leucine zipper (bHLHZip) motif that has DNA-binding activity and can establish protein-protein interactions (*Meyer and Penn, 2008*). Myc regulates a large number of cellular processes, such as proliferation, growth, differentiation and pluripotency, metabolism, and apoptosis (*Eilers and Eisenman, 2008*).

Myc heterodimerizes with another bHLHZip protein, Max, to form the Myc:Max complex that binds to E-box (5′-CACGTG-3′) regulatory DNA elements throughout the genome to control transcription (*Blackwood and Eisenman, 1991*). Myc:Max heterodimers can further form bivalent heterotetramers, allowing them to bring together sequence-distant regions of the genome (*Nair and Burley, 2003*). The heterotetramer assembles in a head-to-tail way through the individual leucine zippers of each heterodimer, resulting in the formation of an antiparallel four-helix bundle. This interaction between Myc and Max is essential for gene transcriptional repression (*Mao et al., 2003*). Several Myc-regulated genes contain multiple E-boxes within promoters that are usually separated by 100 bp (*Coller et al., 2000*). The binding of bivalent Myc:Max to such regions can create loops in the chromatin and allow the cooperative regulation at promoters and enhancers containing multiple E-boxes.

Efforts to model TF binding to chromatin have been made at various spatial levels. At the nucleosome level, all-atom and coarse-grained molecular dynamics simulations have been applied to study the binding of the pioneer TF Oct4, alone (*Huertas et al., 2020*) or with Sox2 (*Tan and Takada, 2020*), to single nucleosomes. At the kb or chromosome level, TF binding has been introduced in polymer models as beads that can reversibly interact with other proteins or with the DNA (*Brackley et al., 2013*; *Buckle et al., 2018*). Although the implicit binding of high-mobility group (HMG) proteins to chromatin fibers has been studied at nucleosome resolution (*Bajpai et al., 2017*), to the best of our knowledge, the effect of TF binding on chromatin architecture has not been studied at nucleosome resolution level.

Here, we perform mesoscale chromatin simulations at nucleosome resolution under the implicit binding of the Myc:Max complex to study this complex's effect on microdomain formation (TAD-like structures at the kb level), its interplay with other chromatin elements (e.g. LH, histone tail acetylation, and linker DNA length), and its effect on gene folding. Overall, we find that depending on the linker DNA length, the TF binding position dictates the folding of fibers and the formation of microdomains. The extent of the effect of TF binding on chromatin compaction and overall shape depends on the presence of LH and histone acetylation, and on the length of linker DNA. While long linker DNAs and high acetylation levels allow higher TF concentrations to further compact the chromatin fibers, short linkers, and LH impose structural restraints that limit the effect of TF binding. Finally, we show that the binding of Myc:Max can repress the expression of the gene Eed by increasing fiber folding and compaction in a way that occludes the promoter region; interestingly, this repression can be reversed with high LH density.

Overall, our study provides new evidence on the regulation of chromatin architecture by TF binding and how such effects are regulated by chromatin composition and epigenetic marks.

## Results
### TF binding mediates the formation of microdomains

Protein binding to chromatin fibers has been shown to produce regions of enriched contacts in Micro-C or Hi-C maps. For example, loops associated with chromosome domain boundaries overlap with CTCF binding regions (*Rao et al., 2014*). Similarly, the enrichment of Hi-C contacts can be used as a reporter of the strength of interaction between a pair of TF binding sites (*Ma et al., 2018*). Finally, the binding of TFs, cofactors, or chromatin modifiers produces fine-scale domains, smaller than TADs, that can be identified as stripes and dots in Micro C maps (*Hsieh et al., 2020*).

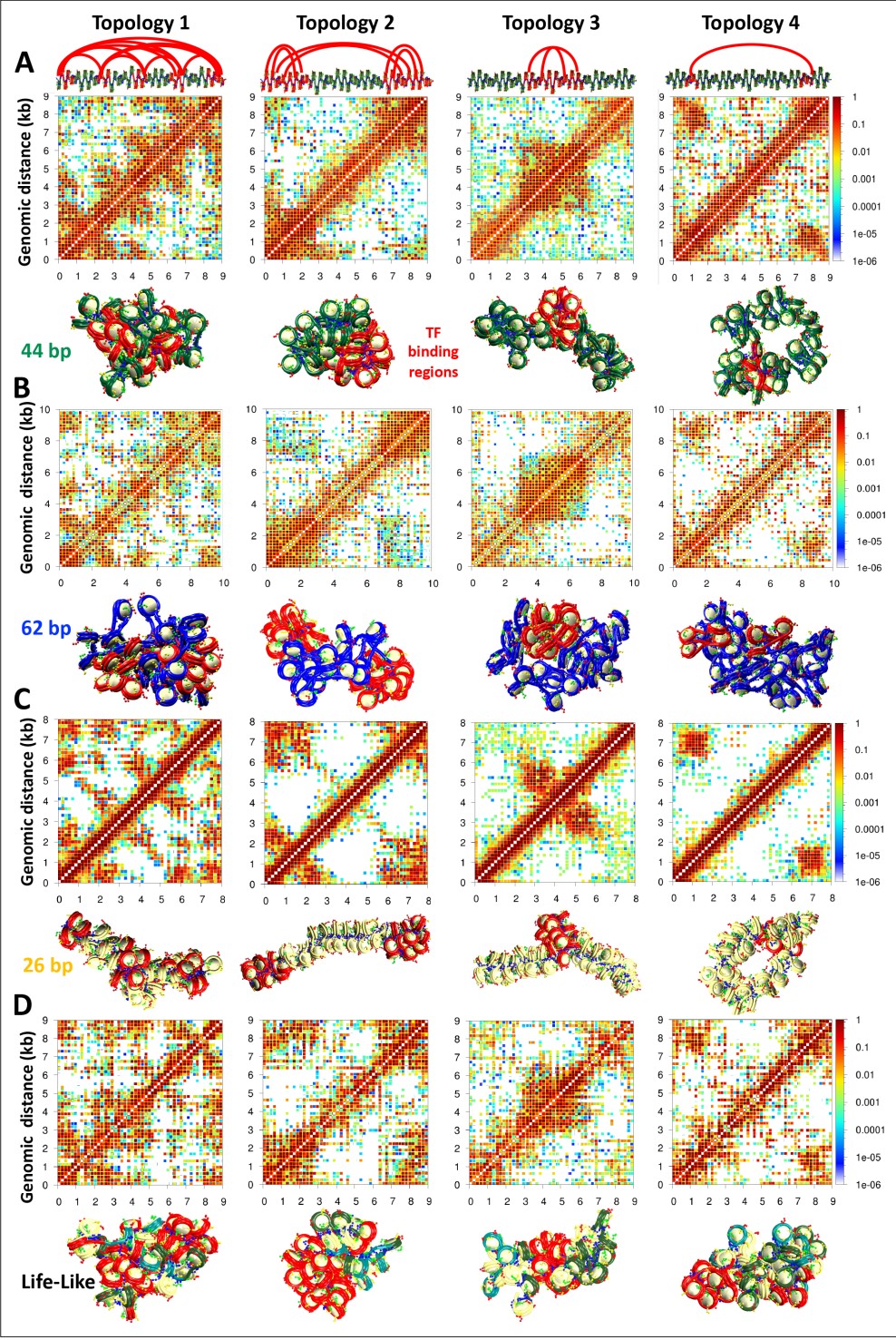

**Figure 1.** Transcription factor (TF) binding location drives the formation of microdomains. 50-nucleosome chromatin fibers with: (**A**) 44 bp linkers, (**B**) 62 bp linkers, (**C**) 26 bp linkers, and (**D**) Nonuniform linkers simulated with four different TF topologies. At the top, for the 44 bp system, we show an ideal zigzag fiber coloring in red the DNA with TF binding regions. Arcs show the possible binding geometries. The binding positions and geometries that define each topology are the same in all systems. For each system, we show the cumulative contact map calculated from 10 independent trajectories and a representative fiber structure also showing in red the TF binding regions. Additional representative structures are shown in *Figure 1—figure supplements 3–6*.

The online version of this article includes the following figure supplement(s) for figure 1:

*Figure 1 continued on next page*

*Figure 1 continued*

**Figure supplement 1.** DBSCAN clustering supports the formation of microdomains.

**Figure supplement 2.** DBSCAN clustering supports the formation of microdomains.

**Figure supplement 3.** Final configurations for each of the 10 independent trajectories of the 44 bp system with four different transcription factor (TF) binding topologies.

**Figure supplement 4.** Final configurations for each of the 10 independent trajectories of the 62 bp system with four different transcription factor (TF) binding topologies.

**Figure supplement 5.** Final configurations for each of the 10 independent trajectories of the life-like system with four different transcription factor (TF) binding topologies.

**Figure supplement 6.** Final configurations for each of the 10 independent trajectories of the 26 bp system with four different transcription factor (TF) binding topologies.

**Figure supplement 7.** Convergence check.

To better understand how TF binding position affects chromatin architecture, we simulate 50-nucleosome uniform fibers of 26, 44, and 62 bp linker DNAs, as well as 50-nucleosome life-like fibers with TF binding in four different scenarios (*Figure 1*). Interestingly, contact maps of fibers with both medium (44 bp) (*Figure 1A*) and long (62 bp) (*Figure 1B*) linker DNAs show clear regions of high-frequency contact, or microdomains, as a result of the TF binding, that are dependent on the TF binding positions. In fibers with short linkers (*Figure 1C*), like 26 bp, the formation of microdomains is less clearly identified in the contact maps. Namely, high-intensity regions close to the diagonal are diffuse and not as well defined as in the fibers with 44 and 62 bp linkers. See for example, topologies 2 and 3 in panels A, B, and C of *Figure 1*. On the other hand, other topologies, like topologies 1 and 4, allow the identification of microdomains. The 44 and 62 bp linker fibers are less compact and more globular than the 26 bp linker fiber (*Perišić et al., 2010*). Additionally, these fibers are more sensitive to changes in salt concentration and the presence of LH (*Perišić et al., 2010*; *Portillo-Ledesma et al., 2022*). The presence of more diffuse microdomains in 26 bp fibers can be explained by the rigidity of short-linker fibers in which the linker DNA dictates fiber architecture and produces a highly bent 10 nm ladder-like form (*Collepardo-Guevara and Schlick, 2014*) with low sensitivity to external and internal parameters, such as LH binding (*Portillo-Ledesma et al., 2022*; *Perišić et al., 2010*; *Routh et al., 2008*). Thus, longer linker DNAs that give chromatin fibers more flexibility, facilitate their folding modulation by TF binding.

Similar to the 44 and 62 bp uniform systems, microdomains emerge in ensemble-based contact maps of life-like fibers (*Figure 1D*). Due to the fiber polymorphism triggered by variations in the linker DNA (*Collepardo-Guevara and Schlick, 2014*), the microdomains are slightly more diffuse than in the 44 and 62 bp uniform fibers (*Figure 1A and B*). Thus, TF binding positions define the organization of life-like chromatin fibers, but the irregularity of the fibers smooths out this effect compared to uniform systems; fibers that combine short and long linkers form fluid heterogeneous bent ladders that accommodate TF binding in a more relaxed way than compact 30 nm regular forms.

As shown in *Figure 1—figure supplements 1 and 2*, these microdomains can also be identified by a clustering analysis of the contact map in nucleosome resolution. Fibers with 44 and 62 bp linkers (*Figure 1—figure supplement 1*) show clear TF-topology-dependent nucleosome clusters. In contrast, for the 26 bp and life-like fibers (*Figure 1—figure supplement 2*), the TF dependency of the nucleosome clusters is not as clear. These results support the notion that microdomains are more diffuse in these two systems due to fiber rigidity or polymorphism, respectively. Similarly, internucleosome contact plots show some TF binding position variations in medium and long-range interactions for the 44 and 62 bp fibers (*Figure 1—figure supplement 1*), but less variations for the 26 bp and life-like systems (*Figure 1—figure supplement 2*). See for example the plots corresponding to Topology 1 of each system where the 44 and 62 bp fibers reveal four clear high-frequency regions at medium and long-range interactions that are not as clear in the 26 bp and life-like systems.

Importantly, the contact maps that reveal microdomains are ensemble-based maps in which the contacts from 10 independent trajectories are summed. These maps are analogous to experimental Hi-C maps obtained from a population of cells. On the other hand, the contact maps corresponding to single trajectories, equivalent to single-cell Hi-C maps, do not show a clear formation of microdomains. As we see from *Figure 2* for the 62 bp system with five TF binding regions (Topology 1),

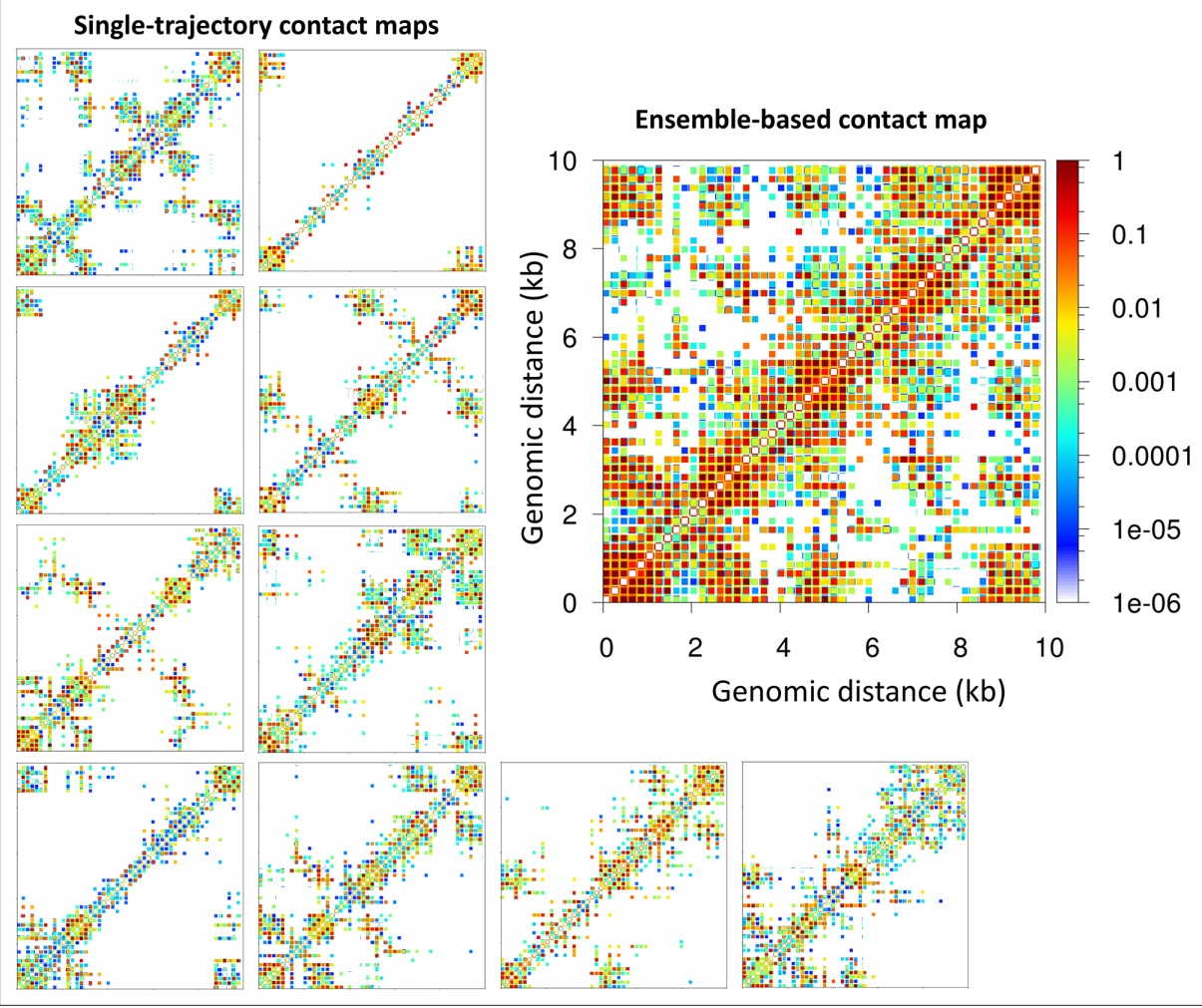

**Figure 2.** Microdomains emerge only from ensemble-based contact maps. The 10 single-trajectory contact maps for the 62 bp system Topology 1 (five Transcription factor (TF) binding regions) at left reveals various microdomains. The large ensemble-based contact map at right obtained by summing the 10 individual contact maps reveals all possible microdomains contacts.

checker-board patterns in the ensemble-based map are a product of TF binding not discernible in individual contact maps obtained from single trajectories. These results agree with experimental data showing that cohesin dictates the preferential positioning of boundaries at CTCF sites to produce ensemble TAD structures (*Bintu et al., 2018*). Here, such microdomain structures emerge naturally from our tagging of TF binding regions.

## TF binding effect is regulated by linker DNA length, histone acetylation, and LH density

As we have shown previously, histone acetylation affects chromatin compaction (*Collepardo-Guevara et al., 2015*), size, and compaction of nucleosome clutches (*Portillo-Ledesma et al., 2021*), and can produce the segregation of domains (*Rao et al., 2017*). Similarly, LH density controls chromatin higher-order folding and compaction (*Portillo-Ledesma et al., 2022*; *Perišić et al., 2019*; *Grigoryev et al., 2016*), and the size and compaction of nucleosome clutches (*Portillo-Ledesma et al., 2021*). In addition, LH binding and tail acetylation act cooperatively to direct fiber folding (*Bascom and Schlick, 2017*).

To investigate how TF binding and other chromatin regulators, such as LH and tail acetylation act together to modulate chromatin architecture, we study TF binding in the presence of LH and tail acetylation for fibers with different linker DNAs. In particular, we determine TF saturation curves for fibers with uniform linkers (26, 35, 44, 53, 62, 70, and 80 bp), and for nonuniform "life-like" fibers (see

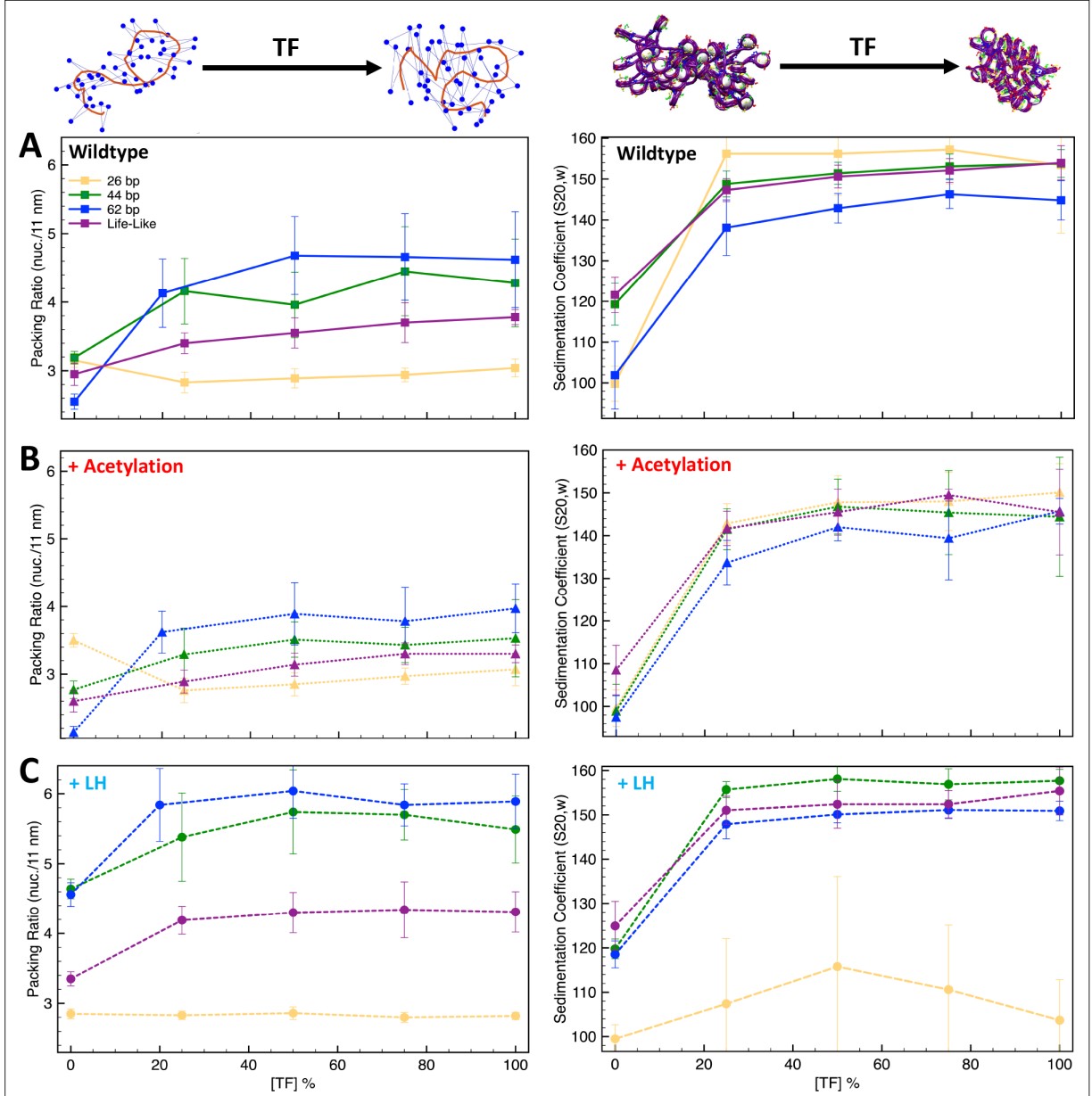

**Figure 3.** Transcription factor (TF) saturation curves are affected by histone acetylation and linker histone (LH). Graphs (**A–C**) show packing ratios and sedimentation coefficients as a function of TF concentration for the 26 bp, 44 bp, 62 bp, and life-like fibers in different conditions: (**A**) systems without LH and acetylation; (**B**) systems with four acetylation islands; and (**C**) systems with LH density $\rho$ = 1. Average and standard deviation values are obtained from ensembles of 1000 structures. At the top left, we show the fiber axis (red trace) and position of nucleosomes (blue dots) for a 70 bp chromatin fiber to illustrate the increase of packing ratio (number of nucleosomes per 11 nm of fiber length) upon TF binding. At the top right, we show a 70 bp linker chromatin fiber to illustrate the decrease of chromatin global size upon TF binding. Results including uniform systems with 35, 53, 70, and 80 bp are shown in *Figure 3—figure supplement 1*.

The online version of this article includes the following figure supplement(s) for figure 3:

**Figure supplement 1.** Transcription factor (TF) saturation curves for the uniform systems of 26, 35, 44, 53, 62, 70, and 80 bp, as well as for the life-like fiber.

Methods) by measuring both packing ratios and sedimentation coefficients at increasing TF concentration (percentage of linker DNA beads that can bind TF), from 0 to 100%.

As we see from *Figure 3A* and *Figure 3—figure supplement 1*, when the linker DNA is short, such as 26 and 35 bp, TF binding does not increase the packing ratio of the fiber. However, the sedimentation coefficient increases significantly from 0 to 25% [TF], and then remains flat. This indicates that TF

binding saturation occurs at a concentration of 25% and that higher TF concentrations do not affect fiber folding. Namely, fibers cannot fold into a more compact form due to excluded volume interactions. As clearly shown by the fiber configurations of the 26 bp system (*Figure 4*), while in the absence of TF the fiber has a ladder-like extended structure (short linkers), it folds over itself when TF is bound, explaining the higher sedimentation coefficients. On the other hand, for medium and long linker fibers, both packing ratios and sedimentation coefficients increase upon TF binding (*Figure 3A*). The longer linkers allow nucleosomes to approach closer to one another upon TF binding, increasing the packing ratio. Additionally, the longer the linker DNA, the larger the change in packing ratio upon TF binding at a concentration of 25%. Similar to the short linker fibers, TF binding also produces a more globular fiber (*Figure 4*) with a higher sedimentation coefficient (*Figure 3A*).

For life-like fibers, the effect on packing ratio and sedimentation coefficient is similar to that observed in the medium and long linker uniform fibers; fibers become more compact and globular upon TF binding (*Figure 3A* and *Figure 4*). However, in agreement with the trends observed for the TF binding regions (*Figure 1*), fluid life-like fibers appear to be less sensitive to the increase of TF concentration than the 44 and 62 bp systems.

When acetylation islands are incorporated into the chromatin systems (*Figure 3B*), the trends in packing ratios and sedimentation coefficients are similar to wildtype systems, but the extent of the TF binding effect is diminished. There is a smaller change in packing ratio when we increase TF binding from 0 to 25%. This is because histone acetylation disrupts internucleosome contacts and opens the chromatin fiber structure (*Collepardo-Guevara et al., 2015*), which is opposite to the repressive effect observed by TF binding. In agreement, compared to wildtype systems, fiber structures (*Figure 4*) show a smaller change in global shape and compaction upon TF binding.

Similarly, when LH is bound to medium and long linker DNA fibers (*Figure 3C*), the effect of TF on chromatin compaction is diminished compared to the wildtype systems. In this case, LH is a chromatin compactor, like TF binding. Thus, LH and TF compete for compacting the chromatin fiber. At 0% TF, the packing ratio of the systems with LH is much higher than the wildtype systems due to LH binding. Thus, upon TF binding, the fiber packing ratio cannot increase as when no LHs are bound. Moreover, LH binding produces straighter and ordered structures (*Portillo-Ledesma et al., 2022*), whereas TF binding produces more globular fibers. Thus, the effects of TF and LH on fiber global shape are in opposition. Upon TF binding, fibers with LH are less globular and more straight than the analogous wildtype fibers (*Figure 4*).

For short linker fibers like 26 and 35 bp, as we saw in the wildtype system, no increase can be observed in the packing ratio upon TF binding. However, very different trends are observed for the sedimentation coefficients of the 26 bp system. In contrast to the wildtype system (*Figure 3A*), TF binding does not increase the sedimentation coefficient (*Figure 3C*), and larger standard deviations are obtained, indicating more variability in the fiber structures that create the 1000-structure ensemble. This is because with LH, some TF dimer/dimer interactions cannot form in some of the trajectories. For example, for the 50% [TF] system, from the 10 independent trajectories, four reach the 50% [TF], four have 0% [TF], and two have less than 20% [TF], producing an effective [TF] in the ensemble of ~22%. Thus, LH bound to short linker DNAs produces a fiber rigidity that impairs TF function. A similar effect has been seen for condensin and topoisomerase II binding (*Choppakatla et al., 2021*), where LH depletion causes excess loading of these proteins to chromosomes, producing aberrant chromosomes.

Finally, for life-like fibers (*Figure 3C*), the trends in packing ratio and sedimentation appear similar to those of the wildtype systems (*Figure 3A*). Interestingly, the increase in packing ratio upon TF binding at a 25% concentration is slightly higher than for the wildtype system (0.85 vs. 0.45 increase). This indicates that the simultaneous binding of LH and TF to life-like fibers is better accommodated than in uniform fibers. Thus, fiber fluidity triggered by variations of linker DNA (*Collepardo-Guevara and Schlick, 2014*) modulates the effect of protein binding on chromatin architecture.

## TF binding-mediated compaction as possible mechanism of gene locus repression

Myc helps maintain cell pluripotency and inhibits differentiation by activating genes needed for pluripotency and repressing genes that trigger differentiation (*Smith et al., 2010*; *Krepelova et al., 2014*). For example, the embryonic ectoderm development gene, Eed, expresses a protein member of the

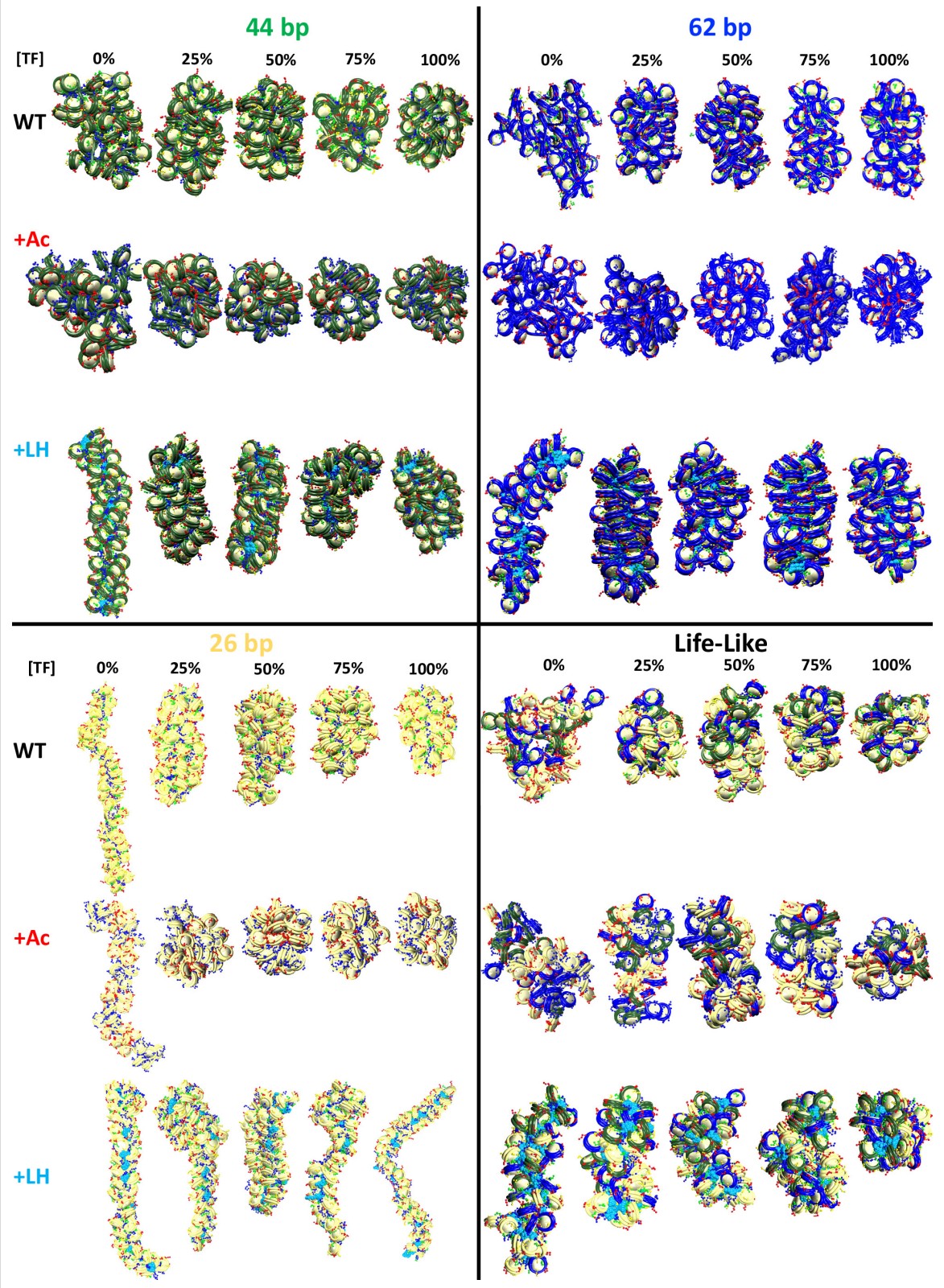

**Figure 4.** Transcription factor (TF) binding affects chromatin architecture. Chromatin uniform fibers of 44, 62, and 26 bp linkers, as well as nonuniform life-like fibers at increasing TF concentration. Wildtype (WT) shows structures of the wildtype systems (no acetylation and no LH). +Ac shows structures of the acetylated systems with acetylated tails drawn in red and wildtype tails in blue. +Linker histone (LH) shows structures of systems with an LH density of 1 LH/nucleosome, with LHs drawn in cyan.

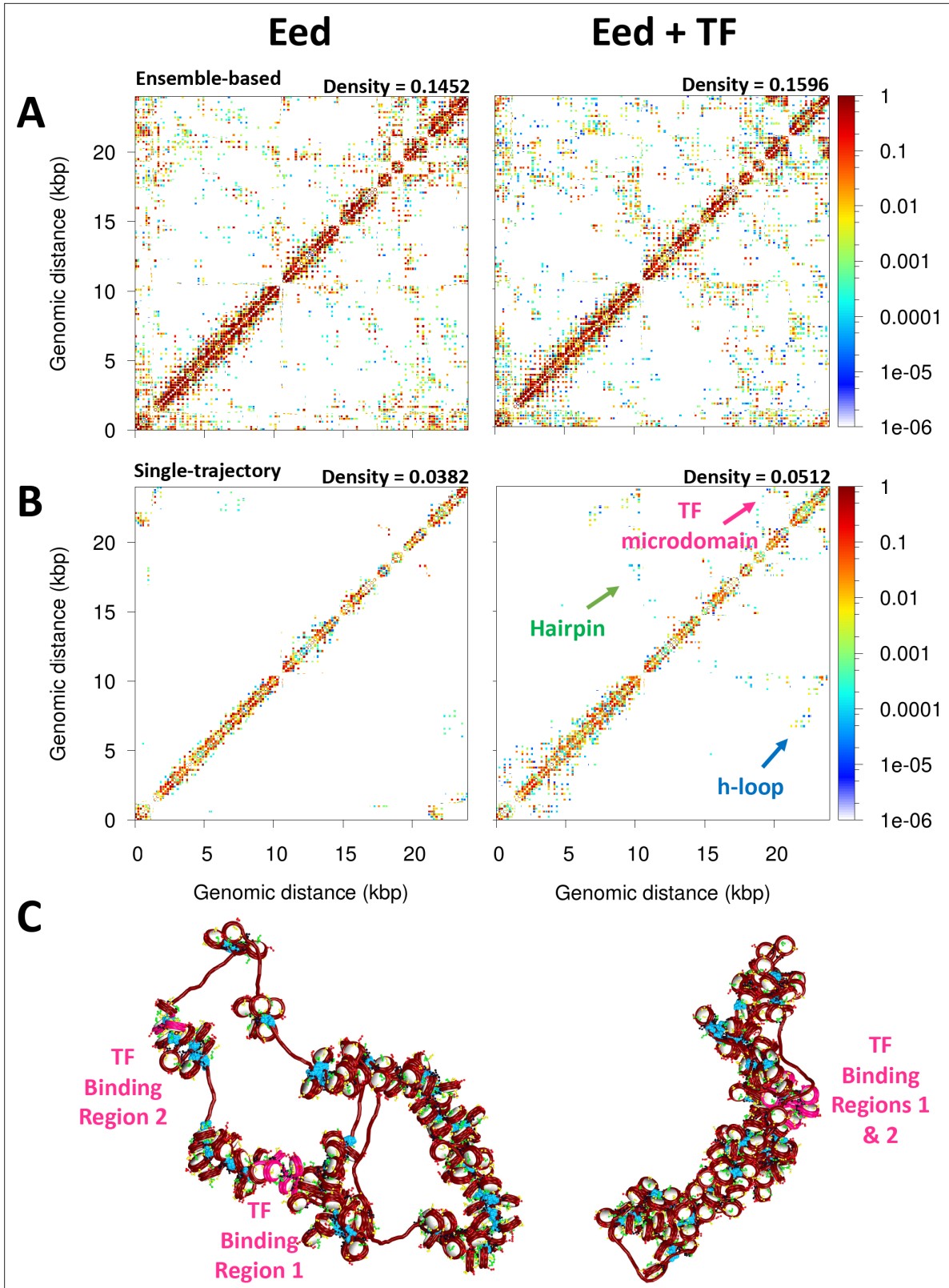

**Figure 5.** Transcription factor (TF) binding represses the Eed gene loci. (**A**) Ensemble-based nucleosome contact maps obtained from 20 independent trajectories of the Eed gene in the absence and presence of TF binding. (**B**) Nucleosome contact maps obtained from a single trajectory of the Eed gene in the absence and presence of TF binding. (**C**) Representative chromatin fibers of the Eed gene in the absence and presence of TF binding. In magenta are shown the TF binding regions. Linker histones (LHs) are shown in cyan.

*Figure 5 continued on next page*

*Figure 5 continued*

The online version of this article includes the following figure supplement(s) for figure 5:

**Figure supplement 1.** Genomic interaction frequency vs. genomic distance for the Eed gene with and without transcription factor (TF) bound.

Polycomb-group (PcG) family required for silencing pluripotency genes upon embryonic stem cell (ESC) differentiation (*Obier et al., 2015*). Thus, in ESCs, Eed is repressed so that cell differentiation is suppressed. To determine the mechanism of gene repression in Eed by Myc:Max, we fold de novo the Eed gene in the presence and absence of TF binding. To study Eed gene activation upon cell differentiation, we additionally simulate the folding of Eed with and without TF binding, and with increasing LH density ($\rho$) to 0.8 LH/nucleosome, the level observed in mouse somatic cells (*Woodcock et al., 2006*).

*Figure 5A* shows the ensemble-based contact map for the Eed gene with and without TF binding. We see that TF binding increases the number and frequency of internucleosome contacts. The density of the matrix increases by ~9% upon TF binding. On the right upper corner of the contact maps, there is a clear difference: microdomains emerge as product of the TF binding. Contact maps obtained from single trajectories (*Figure 5B*) clearly show that TF binding increases hairpin motifs (medium-range contacts) and hierarchical loops (long-range contacts); these motifs are evident from diagonal and perpendicular regions to the main diagonal of the map (*Grigoryev et al., 2016*). Such an increase in local loop interactions is also observed in the plot of contact frequency versus genomic distance (*Figure 5—figure supplement 1*) showing that while short local interactions slightly decrease upon TF binding, local loop interactions increase markedly. These folding motifs increase fiber compaction (*Figure 5C*). Fiber configurations show that upon TF binding, the TF binding regions are in close proximity, producing a more compact fiber.

Furthermore, the fiber sedimentation coefficient increases, while the radius of gyration decreases, upon TF binding (*Table 1*). Such radii of gyration measurements agree with experimentally and theoretically determined radii of gyrations for similar size fibers (*Kadam et al., 2023*; *Boettiger et al., 2016*) (e.g. our 62 ± 14 nm Rg is similar to the 43 ± 9 nm Rg predicted for 26 kb regions in *Drosophila* cells *Boettiger et al., 2016*). Significantly, we find that the TF binding region 1, which overlaps with the promoter region, becomes occluded, thus less accessible. As reported in *Table 1*, the area and volume of the promoter region are reduced ~1.5 times upon TF binding. Thus, Eed mechanism of repression by Myc:Max involves the compaction and occlusion of the promoter region due to increased fiber folding. A less accessible promoter will impair gene transcription.

Upon cell differentiation, LH density increases to ~0.8 LH/nucleosome (*Woodcock et al., 2006*), and Eed needs to be activated so the polycomb protein is expressed and the pluripotency genes are repressed. Thus, to determine how such an LH density increase upon cell differentiation might affect Eed repression and help its activation, we study the Eed gene loci with an LH density $\rho$ = 0.8. When $\rho$ increases, the fiber becomes more compact and less flexible (*Figure 6* and *Table 1*) due to the repressive effect of LH binding (*Portillo-Ledesma et al., 2022*; *Luque et al., 2014*). This change in fiber compaction and flexibility modulates the impact of TF. Similar to the effect observed for the 26 bp system in *Figure 3*, the Myc:Max dimers cannot interact when the fiber becomes more rigid and straight due to LH binding. Indeed, from the 20 trajectories ran for this system, only one trajectory has dimers engaged, and thus, the systems with $\rho$ = 0.8 with and without TF exhibit similar properties (*Table 1*), structures (*Figure 6*), and genomic contacts (*Figure 6—figure supplement 1*). These results implicate LH in the activation of genes by impairing the effect of repressive proteins.

## Discussion

Our Monte Carlo simulations of chromatin fibers at nucleosome resolution with implicitly bound TFs demonstrate how these proteins shape chromatin architecture and thus regulate gene expression. Our results on different TF binding regions (*Figure 1*) show that TF binding creates microdomains that are visible in the ensemble-based contact maps; these microdomains are more clearly evident in, and enhanced by longer-linker DNAs. These results agree with experimental Hi-C maps (*Rao et al., 2014*; *Ma et al., 2018*; *Hsieh et al., 2020*) and polymer modeling studies (*Brackley et al., 2013*; *Brackley et al., 2016*; *Brackley et al., 2017*) showing that the clustering of proteins that bridge genomic segments results in TAD-like structures. Our results showing that fibers with short linkers have more

**Table 1.** Compaction parameters: Sedimentation coefficient and radius of gyration for the entire Eed system, and area and volume for the promoter region of Eed.

|  | Fiber Rg (nm) | Fiber Sc ($S_{20,w}$) | Promoter area ($nm^2$) | Promoter volume ($nm^3$) |
|---|---|---|---|---|
| Eed | 62 ± 14 | 162 ± 19 | 252 ± 45 | 1753 ± 239 |
| Eed + TF | 60 ± 13 | 171 ± 18 | 171 ± 42 | 1262 ± 202 |
| Eed + 0.8 LH | 61 ± 9 | 173 ± 15 | 196 ± 47 | 1447 ±123 |
| Eed + 0.8 LH + TF | 57 ± 8 | 175 ±16 | 210 ± 46 | 1420 ± 104 |

blurred microdomains to geometrical restrictions agree with observations that silent chromatin has longer linker DNA than active chromatin (*Valouev et al., 2011*). Longer linkers provide the chromatin fiber with the flexibility needed to fold and create loops that compact the structure and repress transcription. We showed this for metaphase chromatin where longer linkers and the absence of LH favor the formation of hierarchical loops that compact genes and chromosomes (*Grigoryev et al., 2016*). Importantly, life-like fibers, representative of real-life chromatin, are sensitive to TF binding positions and form microdomains, although these are less defined than in uniform fibers due to the high plasticity and fluidity of the heterogeneous chromatin (*Collepardo-Guevara and Schlick, 2014*).

Similar to the effect on microdomain formation, we also suggest that, upon TF binding, fibers with short linkers (26 and 35 bp) do not change their packing ratio, whereas fibers with medium and long (44–80 bp) linkers, as well as life-like fibers, become more compact (*Figure 3A*). Thus, the role of linker DNA appears to be coupled to the effect that repressive proteins have on the chromatin structure. This agrees with recent cryo-EM results showing that long linker DNAs produce relaxed DNA trajectories that enable the binding of LH and produce transcriptionally silent heterochromatin regions, whereas short linker DNAs preclude the binding of LH and produce transcriptionally active chromatin (*Dombrowski et al., 2022*).

Histone tails extend from the nucleosome core and interact with other nucleosomes and with the linker DNA (*Arya and Schlick, 2006*). When tails are acetylated, they become globular, impairing internucleosome interactions and opening up the chromatin fiber (*Collepardo-Guevara et al., 2015*). When we introduced histone acetylation, we saw that, for fibers with medium, long, and life-like linker DNAs, the effect of TF binding on chromatin compaction is reduced (*Figure 3B*). This is because the effects of acetylation and TF binding (Myc:Max in particular) on chromatin compaction and architecture are opposite. In line with these results, it has been reported that histone acetylation can prevent the binding of some pioneer TFs (*Fuglerud et al., 2018*). Our results thus highlight how histone acetylation can counter and modulate the repressive effect of proteins that bind to chromatin.

When studying the effect of LH binding, we found that LH can impair the effect of TF binding on chromatin architecture (*Figure 3C*). Particularly affected are fibers with short linkers. The C-terminal domain of LH interacts with the linker DNAs to compact chromatin fibers (*Luque et al., 2014*). Thus, LH competes for the linker DNA with TF binding. Indeed, recent experiments have shown that although LH remains bound to the nucleosome dyad upon TF binding, the C-terminal domain unbinds from the linker DNA (*Burge et al., 2022*). These results highlight the influence of the LH C-terminal domain on regulating TF binding. LH binding produces a rigid and straight fiber (*Portillo-Ledesma et al., 2022*) that is not flexible enough to be maneuvered by TF binding. Thus, although TF and LH binding might not be mutually exclusive as they do not share the same genomic binding sites, LH can indirectly impair TF function by affecting the chromatin fiber architecture. Interestingly, this effect seems to be less important in life-like fibers; these fibers better accommodate simultaneous LH and TF binding. Thus, fiber fluidity appears essential for the regulation of chromatin fibers by multiple protein binding.

Our folding of the Eed gene loci upon repression and activation revealed that TF binding produces the formation of microdomains that can occlude the promoter of Eed, repressing its transcription (*Figure 5*). However, this effect is eliminated by increasing the LH density (*Figure 6*), as the fiber flexibility does not allow for the DNA regions with TF bound to come together. Thus, TF and LH binding work together to regulate gene expression. This cooperation further underscores the idea that LH does not always act as a repressor of global transcription. Furthermore, our emphasis of the LH and TF synergy agrees with results found in mESC showing that LH depletion can lead to both an increase

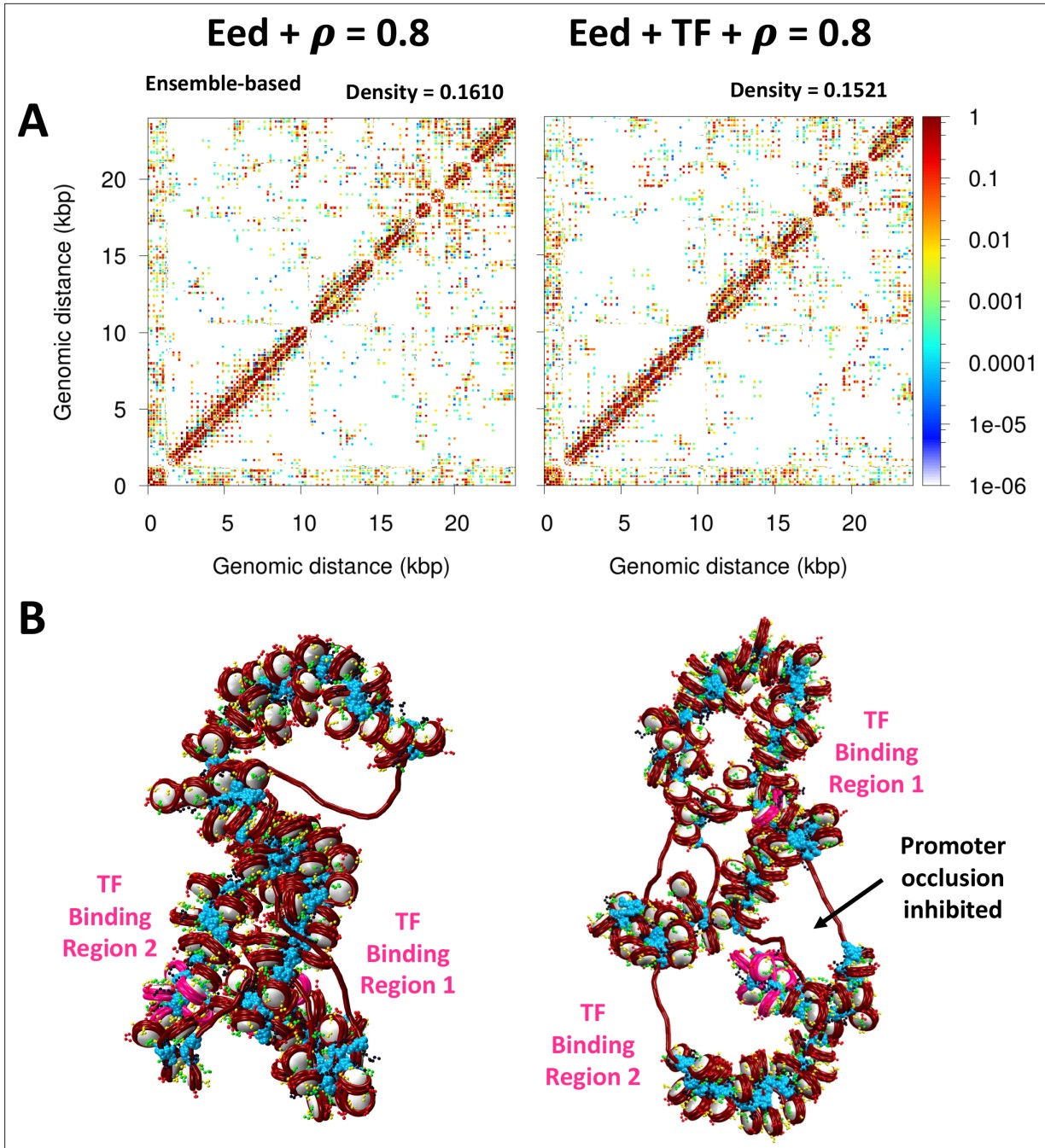

**Figure 6.** Activation of the Eed gene loci depends on linker histone (LH) density. (**A**) Ensemble-based nucleosome contact maps obtained from 20 independent trajectories of the Eed gene with an LH density of 0.8 LH/nucleosome in the absence and presence of transcription factor (TF) binding. (**B**) Representative chromatin fibers of the Eed gene in the absence and presence of TF binding showing that the two TF binding regions remain apart upon TF binding. In magenta are shown the TF binding regions. LHs are shown in cyan.

The online version of this article includes the following figure supplement(s) for figure 6:

**Figure supplement 1.** Genomic interaction frequency vs. genomic distance for the Eed gene with an linker histone (LH) density of 0.8 with and without transcription factor (TF) bound.

and a decrease in gene expression (*Fan et al., 2005*), and that LH depletion in *Drosophila* can downregulate euchromatin genes (*Vujatovic et al., 2012*).

As our nucleosome-resolution modeling study at the gene level reveals, the epigenetic regulation of chromatin by transcription factors is a complex dance that depends on the delicate composition of

other internal and external factors present in the cell. Thus, TFs collaborate with epigenetic marks and intrinsic chromatin features like linker lengths and nucleosome-free regions to regulate the folding of chromatin at multiple scales and hence gene expression.

Because the microdomains and loops affect access to and topologies of transcription start sites, these structural effects translate into effective regulation of cell differentiation and development, as well as human disease progression.

Of course, genome regulation in the live cell involves much more than transcription factors. Other chromatin activators and repressors, architectural proteins, and RNAs are also involved.

Although our implicit modeling of TF binding is a simple strategy, it provides clear trends and guiding insights into the regulation of chromatin architecture by protein binding. Further modeling with explicit proteins or consideration of various biomolecules will undoubtedly increase our understanding of genome regulation on the many structural and temporal levels involved in complex cellular contexts.

## Methods

### Chromatin mesoscale model

Briefly, our chromatin mesoscale model contains coarse-grained elements that define the nucleosome cores, linker DNA, histone core tails, and LH (*Bascom and Schlick, 2017*; *Arya and Schlick, 2006*; *Collepardo-Guevara and Schlick, 2014*; *Perišić et al., 2010*; *Figure 7A*). In particular, cores composed of eight histone proteins and ~147 bp of DNA are described as rigid cylinders with 300 Debye-Hückel charges distributed on their irregular surface calculated by the DiSCO algorithm (*Zhang et al., 2003*). Linker DNA connecting nucleosomes is treated with a combined worm-like chain and bead model (*Jian et al., 1997*) based on the charged colloidal cylinder approach derived by *Stigter, 1977* where each bead has a salt-dependent charge and a resolution of ~9 bp. Histone tails (*Arya and Schlick, 2006*) and LHs (*Luque et al., 2014*; *Perišić et al., 2019*) are coarse-grained as five residues per bead with the Levitt-Warshel united-atom protein model. The charges on histone tails beads are calculated as the sum of the five amino acids that compose the bead, whereas charges on LH beads are determined by the DiSCO algorithm. Acetylation of histone tails is modeled by alternatively folded rigid tails with altered force constants for stretching and bending with energies 100 times larger than wildtype tails (*Collepardo-Guevara et al., 2015*). Tail coordinates are determined separately for wildtype and acetylated tails, by swapping coordinates when a change occurs. The new tail configuration after the swapping is subject to a standard Metropolis acceptance/rejection criterion based on the changes in local electrostatic energy (*Bascom and Schlick, 2017*).

The potential energy function of the model includes stretching, bending, and twisting terms for the linker DNA ($E_S$, $E_B$, $E_T$), stretching and bending terms for histone tails ($E_{tS}$, $E_{tB}$) and LHs ($E_{lhS}$, $E_{lhB}$), and excluded volume ($E_V$) and electrostatic ($E_C$) terms for all beads as:

$$E(\mathbf{r}) = E_S + E_B + E_T + E_{tS} + E_{tB} + E_{lhS} + E_{lhB} + E_V + E_C, \tag{1}$$

where r is the collective position vector.

Further details on the method parameters can be found in *Bascom and Schlick, 2017*; *Arya and Schlick, 2006*; *Collepardo-Guevara and Schlick, 2014*; *Perišić et al., 2010*.

### TF Binding modeling

Based on the crystal structure (PDBID 1NKP) of the Myc:Max complex (*Figure 7B*) showing that heterotetramers can bind to sequence-distant regions, we simulate the binding of Myc:Max implicitly (*Figure 7C*) by adding restraints between two genome loci $b_i$ and $b_j$ (*Figure 7D*). In particular, a harmonic energy penalty of the form:

$$E_{b_i b_j} = k(l_{b_i b_j} - l_0)^2 \tag{2}$$

is applied to two target DNA linker beads. Here $l_0$ is selected as 13 nm based on the distance between the two DNA chains in the crystal structure, and $k$ is set to 20 $kcal/mol\,nm^2$ as this value ensures that the energy penalty is not strong enough to produce overlapping of DNA beads or cores, but small compared to the total energy of the system.

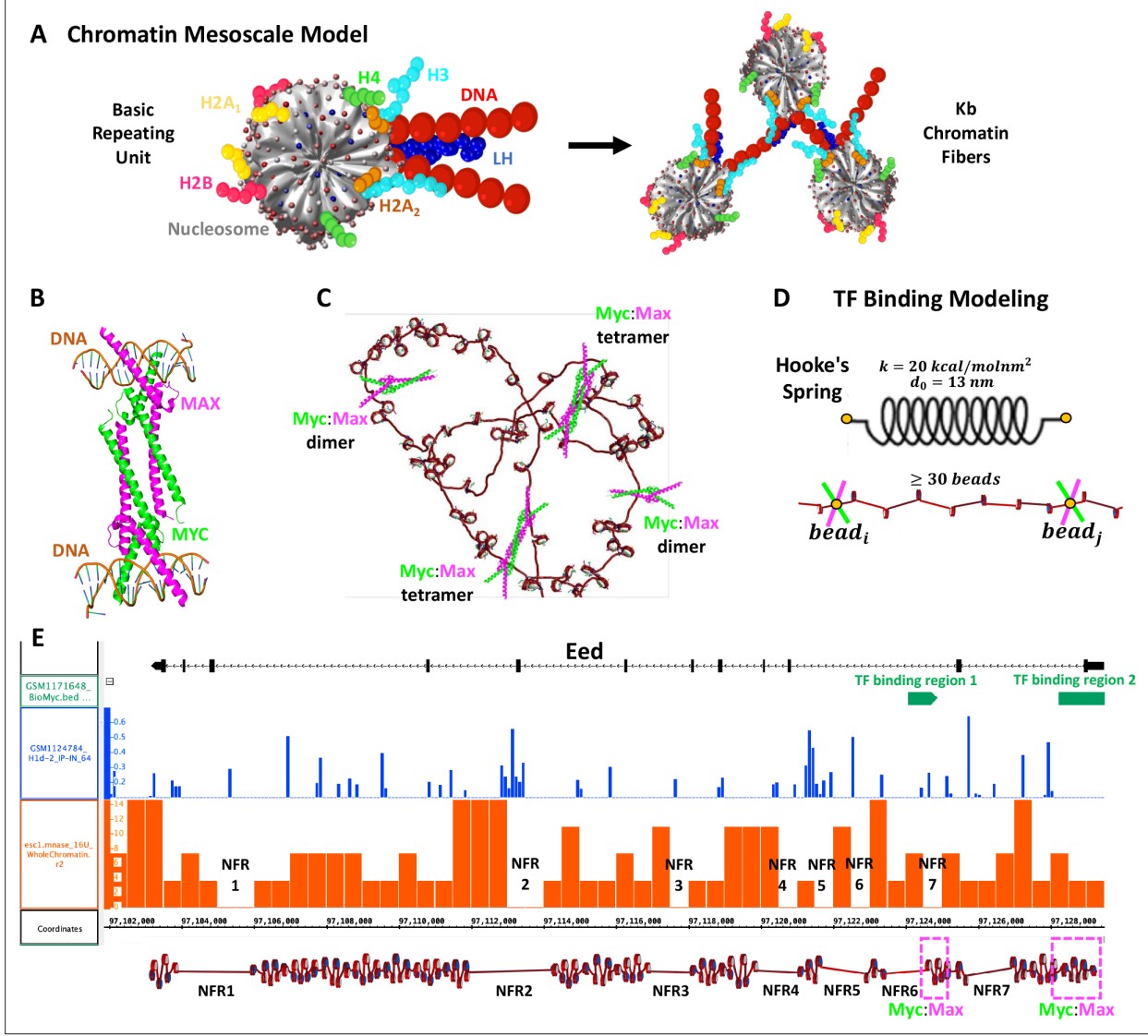

**Figure 7.** Our chromatin mesoscale model and modeling of Myc:Max binding to chromatin and the Eed gene. (**A**) Coarse-grained chromatin elements (nucleosome core, linker DNA, histone tails, and linker histone, LH) form chromatin fibers at the kb level. (**B**) Crystal structure of Myc:Max forming a heterotetramer that binds to sequence-distant DNA regions. (**C**) Schematic representation of Myc:Max heterodimers and heterotetramers. (**D**) Illustration of our implicit modeling of Myc:Max binding to chromatin fibers: distance constraints following Hooke's spring law are enforced when two linker DNAs with Myc:Max bound are in close spatial proximity (20 nm) and at at least 30 linker DNAs apart. (**E**) Design of the Eed gene loci using Mnase-seq data to position nucleosome free regions (NFRs) (orange track) and Chip-seq data to position LHs (blue track) and Myc:Max binding regions (green track).

During each trajectory, two DNA beads that have TF bound can engage in a constraint if the distance between them is less than 20 nm and if they are separated by at least 30 beads. If during the simulation, two beads that were engaged in a constraint separate more than 20 nm, the constraint is eliminated.

## Systems

### TF binding to different regions

To determine the effect of TF binding location on chromatin architecture, we simulate 50-nucleosome fibers without LH and histone acetylation, and with short, medium, or long linker DNA lengths, such as 26, 44, and 62 bp. Additionally, because real-life chromatin fibers are non uniform, we also simulate a life-like fiber with the linker DNAs that follow the distribution of linkers found in mouse embryonic

stem cells (mESCs) (*Voong et al., 2016*): 30% for 26 bp, 17% for 35 bp, 15% for 44 bp, 13% for 53 bp, 9% for 62 bp, 7% for 70 bp, and 9% for 80 bp, as determined in our previous work (*Portillo-Ledesma et al., 2022*).

For each system, we study four different binding patterns (see *Figure 1A*):

1. Topology 1: five equally distributed binding regions that occupy each the linker DNA of 5 consecutive nucleosomes
2. Topology 2: two binding regions, one at the beginning and one at the end of the fiber, that occupy each the linker DNA of 15 consecutive nucleosomes
3. Topology 3: one binding region located in the middle of the fiber that occupies the linker DNA of 15 consecutive nucleosomes
4. Topology 4: two binding regions, one located close to the beginning and another one located close to the end of the fiber, that occupy each the linker DNA of five consecutive nucleosomes

Note that by binding region we mean a region whose linker DNA beads are marked by one Myc:Max dimer.

## TF binding at increasing concentrations

To determine the interplay between TF binding and chromatin internal parameters like LH, histone acetylation, and linker DNA length, we study TF binding at increasing concentrations such as 0, 25, 50, 75, and 100%. In particular, we study 50-nucleosome fibers of 26, 35, 44, 53, 62, 70, and 80 bp linker DNAs, as well as life-like fibers in three conditions:

1. Wildtype, systems have no LH and no acetylation
2. +LH, systems have a density of 1 LH/nucleosome
3. +Acetylation, systems have four histone acetylation islands located at nucleosomes 6–11, 17–22, 28–33, and 39–44

In these three conditions, TFs are allowed to bind to any linker DNA region. TF concentration is calculated as the percentage of linker DNA beads that can bind TF.

## Gene repression by Myc:Max binding

In the mm9 mouse genome assembly, Eed is located on chr7:97,103,164–97,129,486, occupying ~26 kbp. To build Eed, Mnase-seq data (GSM2083107) (*Mieczkowski et al., 2016*) of mESCs were used to position nucleosome-free regions (NFRs). In particular, the data were downloaded in bedGraph format from the Genome Omnibus Expression (GOE) repository and loaded into the UCSC Genome Browser without further processing. NFRs were visually inspected and identified as genomic regions with the absence of signal. Details of MNase-seq data processing can be found at the GEO site where the data are deposited. As shown in *Figure 7E*, 7 NFRs can be identified. NFRs 1 and 2 are long and we model them with ~350 bp, whereas NFRs 3–7 are short, and we model them with ~200 bp. Similar to the life-like fibers, the linker length distribution of mESCs (*Voong et al., 2016*) was used to determine the distribution of nonuniform linker lengths, and the nucleosome repeat length (NRL) typical of mESCs, 189 bp (*Woodcock et al., 2006*), was used to calculate the number of nucleosomes in the 24 kbp fiber (length of the fiber without the NFRs), obtaining 129 nucleosomes. In the Eed gene, two binding regions for Myc:Max have been detected with Chip-seq experiments (GSE48175) (*Krepelova et al., 2014*; *Figure 7E*). One between 97,128,221 and 97,129,793 bp, and a second one between 97,124,070 and 97,124,896 bp. Thus, in the Eed model, these regions are defined as TF binding regions 1 and 2 (*Figure 7E*). Similarly, LH positions were determined based on Chip-seq data (GSE46134) (*Cao et al., 2013*) where peaks with frequencies at least 10% of the highest frequency peak were selected (*Figure 7E*), producing a density of 0.37 LH/nucleosome, close to the average LH density of 0.5 found in mESCs (*Woodcock et al., 2006*). Histone acetylation was not introduced in the model, as analyzed Chip-seq data (*Zhang et al., 2022b*; *Shen et al., 2012*; *Zhang et al., 2022a*) show no acetylation in the Eed region. To determine the role of LH in the Eed activation mechanism, we additionally simulate the Eed system (with and without TF) with LH density $\rho$ = 0.8 (LHs randomly distributed), as found in mouse somatic cells (*Woodcock et al., 2006*). The list of linker DNAs, LH positions, and Myc:Max positions are detailed in Supporting *Table 1*.

## Simulation and analysis

Each system is sampled with Monte Carlo (MC) simulations. For the 50-nucleosome systems, we use 50 million MC steps and 10 independent replicas. For the Eed system of 129 nucleosomes, we use 70 million MC steps and 20 independent replicas. All replicas are started from different random seeds and a DNA twisting angle of 0, +12, or –12 degrees to mimic natural variations (*Drew and Travers, 1985*). Frames are saved every 100,000 MC steps.

The MC moves include a global pivot move that selects the shorter fiber end passing through a random axis and rotates it, local translation and rotation moves of DNA beads and cores, and local translation of LH beads (*Bascom and Schlick, 2017*). Histone tails are sampled with a bead-by-bead regrowth move by the Rosenbluth scheme (*Arya and Schlick, 2006*).

Trajectory convergence is monitored by the system energy and global (end-to-end distance and sedimentation coefficient) and local (nucleosome triplet angle) parameters (*Figure 1—figure supplement 7*) over the entire ensemble (10 trajectories for each 50-nucleosome system and 20 trajectories for the Eed gene). For the 50-nucleosome systems, the last 10 million steps, or 100 structures, of each of the 10 trajectories are used to create ensembles of 1000 structures per system. For the Eed gene systems, we create ensembles of 2000 structures with the last 10 million steps of 20 trajectories. These ensembles are analyzed to calculate fiber packing ratio, sedimentation coefficients, radius of gyration, and volume and area of the promoter region.

The fiber packing ratio is calculated as:

$$P_r = \frac{11 * N_C}{Fl},$$

(3)

where $N_C$ is the total number of nucleosomes and $Fl$ is the fiber length calculated by defining the fiber axis with a cubic smoothing spline interpolation to the nucleosomes $x$, $y$, and $z$ coordinates; see details in the supporting information of *Portillo-Ledesma et al., 2022*.

Sedimentation coefficients are calculated as:

$$S_{20,w} = ((S_1 - S_0) * \rho + S_0) * \left(1 + \frac{R_1}{N_C} \sum_i \sum_j \frac{1}{R_{ij}}\right),$$

(4)

where $S_0$ and $S_1$ are the sedimentation coefficients of a mononucleosome without LH ($S_0 = 11.1$) (*Garcia-Ramirez et al., 1992*) and with LH ($S_1 = 12$) (*Butler and Thomas, 1998*), respectively, $\rho$ is the LH density on the fiber, $R_1$ is the radius of a nucleosome ($R_1 = 5.5$ nm), $N_C$ is the number of nucleosomes in the chromatin fiber, and $R_{ij}$ is the distance between the nucleosomes $i$ and $j$.

The radius of gyration, which describes the overall dimension of the chromatin fiber, is measured as the root mean squared distance of each nucleosome from the center of mass according to:

$$R_g^2 = \frac{1}{N_C} \sum_{j=1}^{N_C} (r_j - r_{mean})^2,$$

(5)

where $N_C$ is the number of nucleosomes, $r_j$ is the center position of the nucleosome core j, and $r_{mean}$ is the average of all core positions.

The volume and area of the Eed promoter region that overlaps with the TF binding region 1 are calculated using the alpha shape function of Matlab. With alphaShape, a bounding area or volume is created to envelop the nucleosome coordinates. We use an alpha value of 100 to create a loose shape. In particular, we use the x, y (for area) or x, y, z (for volume) coordinates of nucleosomes 123–129 as vertices, as they are located in the promoter region.

Interactions among nucleosomes are calculated every 100,000 MC steps for each trajectory, normalized across all trajectory frames, and summed to create a contact map that we plot at a logarithmic scale. Two nucleosomes $i$ and $j$ are considered to be in contact if any element of nucleosome $i$, such as core, tails, or linker DNA, is less than 2 nm from any element of nucleosome $j$. These matrices are calculated in both bp and nucleosome resolution.

Internucleosome interaction matrices at nucleosome resolution are decomposed into one-dimensional plots that depict the magnitude of $i$, $i \pm k$ interactions, or contact patterns, as follows:

$$I(k) = \frac{\sum_{i=1}^{N_C} I'(i, i \pm k)}{\sum_{j=1}^{N_C} I(j)}, \tag{6}$$

where $N_C$ is the number of nucleosomes, $I$ is the internucleosome interaction matrix, and $k$ is the number of nucleosomes between cores $i$ and $j$.

To estimate the number of microdomains in each system, we convert the interaction matrices in nucleosome resolution to distance matrices by calculating the inverse of each element in the matrix ($d = 1/frequency$) and perform a clustering analysis with the DBSCAN algorithm (*Ester et al., 1996*). To define the clusters, we set the parameter *minpoints* as five for every system, and the radius of search, ε, as 3 for the 62 bp system, 2 for the 44 bp and life-like systems, and 1.4 for the 26 bp system.

## Acknowledgements

This work builds upon initial explorations of protein binding to chromatin fibers in our chromatin code by Dr. Gavin Bascom. Support from the National Institutes of Health, National Institute of General Medical Sciences Award R35-GM122562, National Science Foundation Award (2151777) from the Division of Mathematical Sciences, and Philip-Morris USA Inc to TS is gratefully acknowledged. The authors thank the HPC team at NYU for providing computational resources on the Greene NYU Super-computer. SPL is grateful for financial support from the Simons Center for Computational Physical Chemistry at NYU.

## Additional information

### Funding

| Funder | Grant reference number | Author |
|---|---|---|
| National Institute of General Medical Sciences | R35-GM122562 | Tamar Schlick |
| National Science Foundation | 2151777 | Tamar Schlick |
| New York University | | Stephanie Portillo-Ledesma |

The funders had no role in study design, data collection and interpretation, or the decision to submit the work for publication.

### Author contributions

Stephanie Portillo-Ledesma, Conceptualization, Data curation, Software, Formal analysis, Investigation, Visualization, Methodology, Writing - original draft, Writing – review and editing; Suckwoo Chung, Data curation, Formal analysis, Investigation, Visualization; Jill Hoffman, Software, Formal analysis, Investigation; Tamar Schlick, Conceptualization, Resources, Supervision, Funding acquisition, Investigation, Project administration, Writing – review and editing

### Author ORCIDs

Stephanie Portillo-Ledesma ⬦ http://orcid.org/0000-0002-8211-5447
Tamar Schlick ⬦ https://orcid.org/0000-0002-2392-2062

Reviewer #1 (Public Review): https://doi.org/10.7554/eLife.91320.3.sa1
Reviewer #2 (Public Review): https://doi.org/10.7554/eLife.91320.3.sa2
Author Response https://doi.org/10.7554/eLife.91320.3.sa3

## Additional files

### Supplementary files
• Supplementary file 1. EED gene system setup.

• MDAR checklist

## Data availability

Matlab scripts for data analysis, as well as representative structures (in pdb format) for the 26 bp, 44 bp, 62 bp, and life-like fibers, and the Eed system are deposited on Zenodo.

The following dataset was generated:

| Author(s) | Year | Dataset title | Dataset URL | Database and Identifier |
|---|---|---|---|---|
| Portillo-Ledesma S, Chung S, Hoffman J, Schlick T | 2023 | Transcription Factor Binding Regulates Chromatin Architecture | https://doi.org/10.5281/zenodo.8199993 | Zenodo, 10.5281/zenodo.8199993 |

The following previously published datasets were used:

| Author(s) | Year | Dataset title | Dataset URL | Database and Identifier |
|---|---|---|---|---|
| Tolstorukov M | 2016 | ESC1 whole chromatin rep1 16U MNase | https://www.ncbi.nlm.nih.gov/geo/query/acc.cgi?acc=GSM2083107 | NCBI Gene Expression Omnibus, GSM2083107 |
| Cao K, Lailler N, Dong X, Bouhassira EE, Fan Y | 2013 | High Resolution Mapping of H1 Linker Histone Variants in Embryonic Stem Cells | https://www.ncbi.nlm.nih.gov/geo/query/acc.cgi?acc=GSE46134 | NCBI Gene Expression Omnibus, GSE46134 |

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
