## [Editor Report · eLife assessment]

In this **important** study, chromatin is simulated as a polymer at the scale of genes, and the 3D organization of chromatin is analyzed at nucleosome resolution. There is **convincing** evidence for the emergence of chromatin microdomains due to the action of transcription factors, based on the simulation incorporating well-known biophysical properties of DNA, of nucleosomes, of linker histones, and of the transcription factor pair Myc:Max, as well as considering how the 3D organization of chromatin results from bending and looping of DNA. The work greatly improves our understanding of how the joint action of transcription factors and chromatin features affects chromatin structure and accessibility, which is of interest to anyone studying gene regulation.

---

## [Referee Report · Reviewer #1 (Public Review)]

In this study, authors performed multiple sets of mesoscale chromatin simulations at nucleosome resolution to study the effects of TF binding on chromatin structures. Through simulations at various conditions, authors performed systemically analysis to investigate how linker histone, tail acetylation, and linker DNA length can operate together with TFs to regulate chromatin architecture. Using gene Eed as one example, authors found that binding of Myc:Max could repress the gene expression by increasing fiber folding and compaction and this repression can be reversed by the linker histone. Understanding how transcription factors bind to regulatory DNA elements and modulate chromatin structure and accessibility is an essential question in epigenetics. Through modelling of TF binding to chromatin structures at nucleosome levels, authors demonstrated that TF binding could create microdomains that are visible in the ensemble-based contact maps and short DNA linkers prevent the formation microdomains. It has also been shown that tail acetylation and TF binding have opposite effects on chromatin compaction and linker histone can compete for the linker DNA with TF binding to impair the effect of TF binding. This study improves our knowledge on how TFs collaborate with different epigenetic marks and chromatin features to regulate chromatin structure and accessibility, which will be of broad interest to the community.

---

## [Referee Report · Reviewer #2 (Public Review)]

In this paper, Portillo-Ledesma et al. study chromatin organization in the length scale of a gene, simulating the polymer at nucleosome resolution. The authors have presented an extensive simulation study with an excellent model of chromatin. The model has linker DNA and nucleosomes with all relevant interactions (electrostatics, tails, etc). Authors simulate 10 to 26 kb chromatin with varying linker lengths, linker histones (LH), and acetylated tails. The authors then study the effect of a transcription factor (TF) Myc: Max binding. The critical physical feature of the TF in the model is that it binds to the linker region and bends the DNA to make loops/intra-chromatin contacts. Authors systematically investigate the interplay between different variables such as linker DNA length, LH density, and the TF concentration in determining chromatin compaction and 3D organization.

The manuscript is well-written and is a relevant study with many useful results. The biggest strength of the work is the fact that the authors start with a relevant model that incorporates well-known biophysical properties of DNA, nucleosomes, linker histones, and the transcription factor Myc:Max. One of the novel results is the demonstration of how linker lengths play an important role in chromatin compaction (measured by computing packing ratio) in the presence of DNA-bending TFs. As the TF concentration increases, chromatin with short linker lengths does not compact much (only a small change in packing ratio). If the linker lengths are long, a higher percentage of TFs leads to an increase in packing ratio (higher compaction). Authors further show that TFs are able to compact life-like chromatin fiber with linker length taken from a realistic distribution. The authors compute inter-nucleosomal contact maps from their simulated configurations and show that the map has features similar to what is observed in Hi-C/Micro-C experiments. Authors study the compaction of the Eed gene locus and show that TF binding leads to the formation of small domains known as micro-domains. Authors have predicted many relevant and testable quantities. Many of the results agree with known experiments like the formation of the micro-domains. Hence, the conclusions made in this study are justified - they follow from the simulation results.

---

## [Author Response]

The following is the authors’ response to the original reviews.

**Reviewer #1:**

1. The results that TF binding produces microdomains at medium and long linker DNA but not short linker is very interesting. Although the differences can be observed from the figure, it still lacks of quantitative comparison. It is not clear the exact definition of the microdomain observed from simulations and what numbers of microdomains can be identified under different conditions. A quantitative comparison of different conditions could also be provided.

We thank the reviewer for this suggestion. Our intent was to show qualitatively how TF binding locations that we design can direct fiber folding and create microdomains, which we define in the paper as high frequency contact regions in the contact maps, similar to the TADs observed in HiC maps. Together with the fiber configurations, contact maps allow us to identify formation of such microdomains, and to observe how these microdomains change depending on the conditions we build into the model, such as TF binding region or linker DNA length.

To address your point, we have added a clustering analysis of the contact matrices with nucleosome resolution and assign each contact along the genome position (nucleosome index) to a cluster. In Figure 1- figure supplement 1 and 2, we show how DBSCAN clustering provides a clustering distribution that quantitatively describes the microdomains observed in the matrices and estimates the number of microdomains. For example, in the 44 and 62 bp systems, the contacts along the genomic distance separate into 5, 2, and 1 nucleosome groups for topologies 1 to 3, and into 2 and 1 group for topology 4, respectively. In the 26 bp and life-like systems, where microdomains are more diffuse due to fiber rigidity or polymorphism, we see that the clustering results are not as TF-topology-dependent as in the 44 and 62 bp systems. We also decomposed the contact matrices into one dimensional plots that depict the magnitude of *i*, *i* ± *k* internucleosome interactions. We see that internucleosome patterns change with the TF binding topology, and that the 26 bp and life-like systems show the least changes.

1. When increasing TF concentration, from 0 to 100%, it seems that both packing ratio and sedimentation coefficients are not sensitive to the TF concentrations after 25%. Is it due to the saturation of TF binding? How many TF binding sites are considered at each concentration?

Yes, in most cases, at TF concentrations higher than 25%, the fiber compaction does not change due to saturation of TF binding. Although the TF concentrations are reached, such as 50%, 70%, or 100%, these do not influence the fiber architecture. A higher order folding and compaction cannot be reached due to excluded volume interactions that impede overlapping of beads in the model.

We have clarified this in the manuscript.

As stated in the Methods section, the TF concentration refers to the number of linker DNA beads that can engage in a constraint compared to the total number of linker DNA beads. Thus, at 25% TF, 25% of linker DNA beads are engaged in TF constraints. We have added a comment on this in the Results section.

1. It is shown that the contact maps that reveal microdomains are ensemble-based maps and single trajectories do not show clear formation of microdomains. Does the formation of microdomains increase with the number of combined trajectories?

The formation of microdomains occurs in each single trajectory. However, the microdomains formed in each trajectory can be different. That is why ensemble-based maps show clearer trends of microdomains that might not be as visible in single-trajectory maps. If we increase the number of trajectories, the macrodomains will be more visible and there will be more macrodomains in the contact map, but the formation of microdomains will not increase in each single trajectory.

1. "As we see from Figure 3A, when the linker DNA is short, such as 26 and 35 bp, TF binding does not increase the packing ratio of the fiber." The results of 35bp cannot be found in Figure 3A. In addition, the color of 44 and 62 bp should be changed since they are very similar in the figure.

Thank you for catching this. The results corresponding to the 35 bp system are presented in the Figure 3-figure supplement 1. We have changed the text to read “As we see from Figure 3A and Figure3- figure supplement 1..”.

We have changed the color of the 62 bp trace to blue in the plots of Figure 3. Consistently, we have also changed the color of the 62 bp fiber in Figure 1 and Figure 4.

1. For modelling of TF binding at increasing concentrations, it is mentioned that in these three conditions, TFs are allowed to bind to any region. Do you mean TF can also bind to nucleosomal DNA? Nucleosome structure prevents the binding of many TFs.

In our model, only linker DNA beads can engage in the constraints (bind TF).

We have changed the text to read “TFs are allowed to bind to any linker DNA region”.

1. The details of the Mnase-seq dataset and how NFRs are identified should be provided, such as the coverage of the data and what read fragments are selected for NFR mapping.

MNase data in bedgraph format were downloaded from the Genome Expression Omnibus (GSM2083107) repository and loaded without further processing into the Genome Browser. NFRs were visually inspected and detected as genomic regions without peaks. As detailed in the GEO repository, the sequenced paired-end reads were mapped to the mm9 genome. Only uniquely mapped reads with no more than two mismatches were retained and reads with insert sizes less than 50 or larger than 500 bp were discarded.

We have clarified this in the manuscript.

1. The calculations of volume and area of the Eed promoter region should be further elucidated.

Thank you. We now elaborate upon these calculations. In particular, the Eed promoter region is defined between cores 123 and 129. The x,y or x,y,z coordinates of those cores are used to create the bounding area or volume by defining the shape’s vertices.

1. In Figure 2, it is not clear how different topology are identified.

In Figure 2 the topology, or TF binding regions, is the same for each of the 10 contact maps as these emerge from trajectory replicas of the same system which we named Topology 1. Different microdomains are formed in each individual trajectory as the high-frequency regions appear in different locations on each contact map. However, when these 10 maps are summed, the ensemble contact map clearly shows consensus microdomains in each region where TF binds.

**Reviewer #2:**
To further improve the manuscript, I have the following suggestions/comments.1. While most of the conclusions in this paper follow from the evidence provided by the ximulations, the result in section 3.3 title "Gene locus repression is medicated by TF finding," may not follow from the results. In my opinion, repression is a more complex process, and many more factors (such as nucleosome positioning, nucleosome sliding, histone methylation, and other proteins such as PRC or HP1, etc) may be involved in repression. While compaction is often associated with repressed chromatin (heterochromatin), recent studies have shown that heterochromatin fibers are highly diverse, and compaction alone may not be the criteria for repression (eg. see Spracklin et al. Nat. Struct. Mol. Biol. 30, 38-51 (2023).). In this light, I would recommend slightly modifying the title to say, "TF binding-mediated compaction can help in gene locus repression" or something similar.

Yes! We completely agree that gene repression is a very complex phenomenon that involves many factors that we are approaching by modeling starting from the simplest strategy. Thus, we have changed the subtitle to read “TF binding-mediated compaction as possible mechanism of gene locus repression”.

1. Authors could also present the contact probability versus genomic distance. This may provide some generic features at nucleosome resolution, given the variability in linker length and LH density.

We thank the reviewer for this suggestion. We have now calculated the contact probability for the EED gene with and without TF binding (Figure 5- figure supplement 1). We see that the contact probability corresponding to short range interactions (i ± 2, 3, 4, 5, and 6) is slightly lower for the EED gene upon TF binding. However, a striking increase in the contact probability upon TF binding is seen in the genomic region between 3 and 5 kb, which corresponds to local loop interactions. Thus, TF binding slightly decreases local interactions but increases chromatin loops. Such changes are not observed for the EED system with LH density 0.8 (Figure 6- figure supplement 1), further supporting the idea that an increase in LH density hampers the effect of TF binding for the EED gene architecture.

We have now added these results to the manuscript.

1. Write a short paragraph about the limitations of the model/study. For example, one of the limitations could be that, as of now, it has only the effect of a few proteins, but to predict repression, one may need to incorporate the effect of several proteins.

We agree with the reviewer that our model is a simple, first-step approach. Nonetheless, even the simplest mathematical model can be enlightening in helping dissect essential factors. Here, our model clearly shows how TF binding location modulates fiber architecture and the interplay between TF binding and other chromatin elements, like linker DNA length, LH density, and histone acetylation. We have now stated in the Discussion section that although limited due to being implicit and not considering other protein partners, our model can provide insights on the regulation of chromatin architecture by protein binding. Future modeling with explicit protein binding or combination of several proteins will further help us understand genome folding regulation.

1. The radius of gyration of 26 kb chromatin is around ~60nm in this paper. Is there any experimental measurement to compare (approximate order of magnitude)? While I do not know any measurement for Eed gene locus, I am aware of the results in the Boettiger et al. paper from Xiaowei Zhuang lab (Nature 2016). There, they find that the Rg of a 26 kb region is above 100nm. But that is for a different organism, a different set of genes. Also, see Sangram Kadam et al. Nature Communications 14 (1), 4108, 2023.

Thank you for this suggestion. To the best of our knowledge, there are no radius of gyration measurements for the EED gene. Regarding the two papers you cite, in the paper from Boettiger et al. (1) they determine by microscopy experiments that Rg ∝ *L*! where *L* is the genomic length and *c* is 0.37 ± 0.02 for active chromatin (Figure 1d of the paper). In such case, the Rg for a 26 kb region would be 43 ± 9 nm. Considering that these are *Drosophila* cells, our value of 62 nm is in good agreement with that estimate. Regarding the Kadam et al. paper (2), by coarse grained modeling they find an Rg of around 100 nm for different genes. Considering that the radius of gyration depends on cell type and fiber configuration (see for example (3) for the dependency of Rg on loop number and persistence length), we believe that our measurements in the same ball park as experimental results and other theoretical modeling studies are good indicators of our model’s reasonableness.

We have added this comparison to the manuscript.

1. The reason why it is useful to compare some distance measurements (physical dimension) with experiments is the following: The contact map in Hi-C only gives relative contact probabilities. It does not give absolute contact probabilities. To convert a Hi-C map into a physical distance, one requires comparison with some experimentally measured 3D distance. The radius of gyration is an ideal quantity to compare. From my experience, the contact probability is often much smaller than 1, suggesting that the chromatin is more expanded. But this could be due to the effect of many other proteins in vivo and the crowding, etc. I do not expect this work to incorporate all those effects. However, it may be useful to make a comment about it in the manuscript.

Thank you. We have added to the discussion a comment on our first-generation model of TF binding to chromatin and the neglect of many associated protein and RNA cofactors that certainly influence chromosome folding and domain formation on higher scales. Some distance measures are also added to the Results as mentioned above.

References

1. Boettiger,A.N., Bintu,B., Moffitt,J.R., Wang,S., Beliveau,B.J., Fudenberg,G., Imakaev,M., Mirny,L.A., Wu,C. and Zhuang,X. (2016) Super-resolution imaging reveals distinct chromatin folding for different epigenetic states. Nature, 529, 418–422.

2. Kadam,S., Kumari,K., Manivannan,V., Dutta,S., Mitra,M.K. and Padinhateeri,R. (2023) Predicting scale-dependent chromatin polymer properties from systematic coarsegraining. Nat. Commun., 14, 4108.

3. Wachsmuth,M., Knoch,T.A. and Rippe,K. (2016) Dynamic properties of independent chromatin domains measured by correlation spectroscopy in living cells. Epigenetics Chromatin, 9, 57.